# Comparison of Polymerase Chain Reaction and Urine Culture in the Evaluation of Patients with Complex Urinary Tract Infections

**DOI:** 10.3390/biology13040257

**Published:** 2024-04-13

**Authors:** Deepak A. Kapoor, Mara R. Holton, Jason Hafron, Rima Aljundi, Bernadette Zwaans, Mitchell Hollander

**Affiliations:** 1Solaris Health Holdings, LLC, Fort Lauderdale, FL 33394, USA; 2Anne Arundel Urology, Annapolis, MD 21401, USA; mholton@aaurology.com; 3Michigan Institute of Urology, St. Clair Shores, MI 48081, USA; hafronj@michiganurology.com (J.H.); aljundir@michiganurology.com (R.A.); hollanderm@michiganurology.com (M.H.); 4Corewell Health William Beaumont University Hospital, Royal Oak, MI 48073, USA; bernadette.zwaans@corewellhealth.org

**Keywords:** urinary tract infection, urine culture, polymerase chain reaction, complex UTI

## Abstract

**Simple Summary:**

Complex urinary tract infections are difficult to treat, consume significant resources, and have an increased risk of causing harm to the patient. Conventional urine cultures (UCs) have been shown to be inadequate in diagnosing the causes of complex urinary tract infections. We sought to compare molecular urinary testing using polymerase chain reaction (PCR) and UC in patients with complex tract infections in a single practice. We found that in 3395 patients seen in a single year, PCR detected 36.4% more organisms than UC. PCR identified 91.0% of the organisms found in UC, while UC identified 40.7% of the organisms identified by PCR. In total, 62.4% of the organisms were identified by PCR alone, while 9.4% of the organisms were found only by UC. All these findings were statistically significant (*p* < 0.05). Our findings indicate that in this high-risk, difficult-to-treat subset of patients, while PCR is superior to UC in the identification of potentially causative organisms, urine culture should not be completely replaced by PCR. Ideally, these tests should be used in combination to provide clinicians with the best possible information on which to base their clinical decision making.

**Abstract:**

To compare organism identification using polymerase chain reaction (PCR) and urine culture (UC) in patients with complex urinary tract infections (cUTIs), we reviewed the results of 3395 patients seen during 2022 with cUTI who underwent concomitant PCR and UC testing. We compared the overall positivity rates as well as the ability of each test to identify fastidious organisms (FOs) and the presence of polymicrobial infections (PMOs) and conducted concordance analysis between the tests. PCR detected 36.4% more organisms than UC and was 20 and nearly 36 times more likely to detect PMOs and FOs, respectively. PCR identified 90.6% of organisms found in UC, whereas UC identified 40.7% of organisms found in PCR testing. We found that 62.4% of organisms found in PCR were not found in urine culture, while UC found 9.4% of organisms not identified in polymerase chain reaction. All these differences were statistically significant (*p* < 0.05). Although we found that PCR was superior to UC in overall pathogen detection, and detection of both PMOs and FOs, both identified potentially pathogenic organisms not found in the corresponding test. Our data strongly suggest that the evaluation of patients with cUTI is best accomplished using PCR in conjunction with UC.

## 1. Introduction

Urinary tract infections (UTIs) are a leading cause of patient visits to healthcare providers, accounting for approximately 150 million cases annually worldwide [1]. This includes one million emergency department visits and over 626,000 hospitalizations, with an estimated annual healthcare spending of USD 3.5 billion in the United States alone [2]. While simple UTI rapidly resolves with treatment, a subset of patients present with or develop complicated UTI (cUTI), which is more likely to be associated with adverse outcomes, including a protracted therapeutic course, increased morbidity, and even mortality [3]. Because of the potential for serious and long-term complications and adverse effects on the GU tract or other organ systems in cUTI, it is crucial that pathogen identification is both rapid and correct [4].

Traditional urine culture (UC) has been the cornerstone of pathogen identification in UTI for over a century. UC is challenged by time and technical limitations and requires both selective and non-selective media and technical personnel to interpret the results [5,6]. Tradition diagnosis of UTI requires a growth threshold of 10^5^ CFU/mL, although the guidelines have been modified to lower the threshold value based on patient history, presentation, and other factors [7]. False-negative UC results may also occur due to various technical reasons [8]. In addition, pre-analytic factors such as faulty specimen handling may result in inappropriate false-positive or false-negative urine culture results [9]. While the literature suggests that *E. coli* is responsible for up to 80 percent of uncomplicated UTIs, reports suggest that this organism is responsible for as little as one-third of cUTIs [2,10]. In addition, *E. coli* grows rapidly in standard culture media and can competitively suppress coincident pathogens, leading to non-detection of the organism. Recently, several pioneering studies have suggested that there may be substantial improvements in identifying polymicrobial pathogens using syndromic molecular assays with polymerase chain reaction (PCR) technology, [11,12,13] and in addition, that PCR testing is best accomplished by a relatively expanded panel size [14]. While important, the literature is limited either due to a relatively small number of patients, or in the case of multi-center studies, the possibility that there was variance between sites in either patient selection, PCR protocols, or both.

In our multisite urology practice, we developed practice guidelines that govern the use of PCR testing and sought to ascertain whether there was a difference in overall pathogen detection between PCR and conventional UC testing, specifically in patients who met these guidelines. Furthermore, we sought to evaluate whether this difference, if present, extends to the identification of fastidious organisms (FOs) or UTIs where polymicrobial organisms (PMOs) are present. To further assess the clinical utility on an individual specimen basis, we performed a line-item analysis for each sample to determine whether PCR or UC uniquely identified potentially pathogenic organisms not found in the other test.

## 2. Materials and Methods

### 2.1. Study Design

In our multi-center single-institution independent practice of urology, we restrict the use of urinary molecular testing to those patients who meet clinical parameters supporting the diagnosis of cUTI; these criteria are summarized in Table 1. We do not perform PCR testing on patients with indwelling urethral catheters. We reviewed the laboratory data of all patients greater than 18 years of age seen in our institution during the calendar year 2022, who met these practice criteria and had concomitant UC. The study was conducted in accordance with the Declaration of Helsinki and was reviewed and determined to be exempt from IRB approval (IRB 2023-022).

### 2.2. Specimen Collection and Processing

Each patient’s urine was collected through either midstream clean catch or indwelling urinary catheter in a 4.0 mL gray-top BD Vacutainer^®^ Plus C&S Boric Acid Sodium Borate/Formate (Becton, Dickinson and Co., Franklin Lakes, NJ, USA) tubes specifically designed to transport urine with minimal sample degradation. The protocol for urine culture collection is provided in Appendix A. All urine specimens were shipped to P4 Diagnostix (Pine Brook, NJ, USA) via an overnight service. Once received, the urine specimens were first accessioned and then processed for UC and subsequently PCR.

### 2.3. UC and PCR Laboratory Analysis

Urine culture and PCR were performed in accordance with the published methodology of the reference laboratory [14], and additional information on specimen processing is provided in Appendix A. In brief, urine culture was performed by plating 1 µL urine onto blood–MacConkey agar (ThermoFisher Scientific, Waltham, MA, USA) and incubated at 35–37 °C for at least 18 h. The plates were then examined for colony growth, quantity, and morphology. Microbial growth and sensitivity were assessed using Vitek2 (bioMérieux SA, Marcy-l’Étoile, France). Cultures with no visible growth were re-evaluated after an additional 24 h. Positive UC (UC+) was defined as any single organism with a growth of >10,000 colony-forming units (CFU)/mL. For DNA extraction, nucleic acid isolation was performed on 800 µL of patient sample using MagMax Ultra kit (ThermoFisher Scientific, Waltham, MA, USA) and enzyme mix for lysis and run on the Kingfisher Flex platform. PCR analysis was performed by separately adding 2.5 µL of DNA and 2.5 µL of master mix into a 384-well plate and then spotting onto an OpenArray™ Chip (ThermoFisher Scientific, Waltham, MA, USA) in duplicate for each sample and loading onto the QuantStudio™ 12 K Flex Real-Time PCR System (ThermoFisher Scientific, Waltham, MA, USA) for DNA amplification. A total of 36 organisms (Table 2) and 18 antibiotic resistance genes (Table 3) were included on the UTI OpenArray panel. FOs were identified by P4 Diagnostix and are denoted by * in the table. Positive PCR (PCR+) was defined as any organism displaying a cycle threshold < 29.75.

### 2.4. Data Analysis

We determined the overall positivity rates and identified the PMOs and FOs for each test, stratified by collection methodology (clean catch vs. straight catheter). We performed concordance testing at both the organism and specimen levels to assess whether each test identified unique potentially clinically actionable data. Overall, organism concordance was defined as the identification of any individual organism in both the tests. Specimen concordance was defined as a complete match between all the organisms identified in both tests. This analysis was performed for all organisms and was stratified by the presence of PMOs and FOs. For PMOs, concordance was defined after Hao et al. [14] as the same number and type of organism in both tests (for example, if two organisms were present in UC, to be classified as concordant with PCR, the same two and only two organisms would need to be present in PCR). Statistical analysis of specimen positivity was performed using the chi-square test while differences between clean-catch and catheterized specimens were determined using Fisher’s exact test. Individual organism line-item concordance was analyzed using Student’s paired *t*-test and treatment modifications based on empiric therapy and PCR results determined by a two-proportion z-test. Statistical analysis was performed using GraphPad Prism version 10.2.0 (GraphPad Software, San Diego, CA, USA) and Microsoft Excel (Microsoft^®^ Excel^®^ for Microsoft 365 MSO (Version 2307 Build 16.0.16626.20170) 64-bit).

## 3. Results

The age and sex distributions of the patients are shown in Figure 1. Of the 3395 patients who met the inclusion criteria, 53.3% (1808) were female and 46.7% (1587) were male (D6.6%%, *p* < 0.01). Most patients (78.9%, *p* < 0.01) were older than 60 years, and gender differences were influenced by age. For patients younger than 60 years, our population included 63.5% women and 36.5% men (D26.9%, *p* < 0.01); however, for those older than 60 years, the gender difference in our population was nominal (50.5% female, 49.5% male, D1.0%, *p* = 0.96). In total, 66/3395 (1.9%) patients had specimens collected via straight catheter (34 males, 32 females). As per practice protocol, no patient who underwent PCR testing had their specimen collected via indwelling catheter.

The overall test results are shown in Figure 2. Based on PCR, 95.3% of the positive urine cultures were also identified as positive. Of the positive UCs that were determined to be negative based on PCR, 48.6% (36/74) were due to organisms that were not present on the PCR panel. More than one-third (35.2%) of the UC samples tested positive by PCR (*p* < 0.01). Of the UC specimens, 46.1% (1565/3395) and 62.8% (2135/3395) were positive by PCR (16.8%; *p* < 0.01).

We also reviewed the UC and PCR results based on the method of collection (Table 4). We observed that in both UC and PCR, test positivity was higher in catheterized vs. clean-catch collections. For PCR, 49/66 (74.2%) vs. 2086/3329 (62.7%) catheterized and clean-catch samples, respectively, were positive (difference: 11.6%, *p* = 0.07). For UC, 38/66 (57.6%) vs. 1527/3329 (45.9%) samples were positive (difference: 11.7%, *p* = 0.08). 

We also reviewed the difference between positive PCR and UC test results and the results of the corresponding test in patients that were and were not catheterized, as summarized in Table 5. We found that in PCR-positive clean-catch samples, 1442/2025 (70.3%) UCs were positive, while 36/49 (73.5%) catheterized PCR-positive specimens were UC-positive (difference: 3.2%, *p* = 0.75). Conversely, in positive UCs, 1455/1527 (95.3%) and 36/38 (94.7%) samples obtained by clean catch and catheter, respectively, were PCR-positive (difference: 0.6%, *p* = 0.70).

We found that PCR was significantly more likely to detect both PMOs and FOs than UC was. With respect to multiple organisms, PMOs were found in 2.1% (33/1565) of UC+ specimens versus 42.0% (896/2135) of PCR+ specimens (difference: 39.9%, *p* < 0.01). All 33 PMOs found in the UC+ specimens grew into two organisms. Of the 896 PMOs identified by PCR, 63.5% (547) had 2 organisms, 26.6% (229) had 4 or more organisms, and 13.9% (120) had 4 or more organisms. When stratified by specimen collection methodology, we found that 20/49 (40.8%) and 876/2086 (42.0%) positive PCR results obtained by catheterization and clean catch, respectively, had PMOs (difference: 1.2%, *p* = 0.99). Simultaneously, we found that 1/38 (2.6%) and 32/1527 (2.1%) positive UC results obtained by catheterization and clean catch, respectively, grew multiple organisms (difference: 0.5%, *p* = 0.56).

Our analysis showed that FOs were found in 0.6% (9/1565) of UC+ specimens and 21.4% (456/2135) of PCR+ specimens (difference: 21.4%; *p* < 0.01). To assess the overlap between the PMOs and FOs in our patients with cUTI, we also evaluated the frequency with which patients tested by UC or PCR had either PMOs or FOs, and whether each test had both PMOs and FOs. For UC+ specimens, 2.7% (42/1565) showed either PMOs or FOs, compared with 46.8% (999/2135) for PCR+ specimens (difference: 44.1%, *p* < 0.01). In total, 16.5% (353/2135) of the PCR+ specimens had both PMOs and FOs; no UC+ specimens had both PMOs and FOs. 

Individual organism counts, line-item organism concordance between the tests, and statistical analysis are shown in Table 6. In total, 3571 instances of 50 unique organisms were identified in one or both tests: 34 (68%) were identified by PCR, whereas 41 (82%) were isolated in UC (difference: 14.0%, *p* = 0.11). In total, there were 1454 instances of 25 unique organisms showing line-item concordance (LIC); this represented 40.7% of the organisms detected by PCR and 91.0% of the organisms identified by UC (difference: 50.3%, *p* < 0.01). The percentage of LIC to total organism count for each test is also depicted in Table 3. In total, 1595 UC+ specimens showed 1598 incidences of 41 organisms, whereas 2135 PCR+ samples identified 3571 instances of 34 organisms. Of the 41 organisms grown in UC, 39.0% were not identified by PCR, whereas 26.5% of the organisms identified by PCR were not isolated from UC (difference: 12.5%, *p* = 0.25). There were 36 instances of the 16 organisms found in UC but not PCR, which represented 2.3% of the 1598 organisms found in UC; conversely, there were 666 instances of the 9 organisms found by PCR but not UC, which was 18.7% of the 3571 organisms identified by PCR (difference: 16.4%, *p* < 0.01). Only 5.6% of the organisms in the PCR panel were not detected in any of the samples (*C. urealyticum* and *M. tuberculosis*).

The presence of organisms uniquely identified by UC and PCR is shown in Table 7: 9.4% (147/1565) of UC+ and 62.4% (1333/2135) of PCR+ specimens had organisms that were not found in the other tests (difference: 53.0%, *p* < 0.01). The specimens were further stratified according to the presence of one or more identified organisms: 36.8% (456/1239) of single-organism PCR+ samples revealed organisms not identified in UC; 7.9% (36/456) of these were UC+ for other organisms. A total of 8.4% (128/1532) single-organism UC+ samples were not detected by PCR (difference: 28.2%, *p* < 0.01); of these, 43.4% (53/128) were PCR+ for organisms not identified in UC. Of the 128 samples, 30 (23.4%) were organisms that were not on the PCR panel. 

## 4. Discussion

Urinary molecular testing using either PCR amplification or next-generation sequencing has identified a multitude of organisms in the ‘healthy’ bladder, which challenges conventional wisdom that urine is sterile at baseline [15]. It is now clear that the historical notion of sterility in urine is at least partially predicated on limitations of urine culture. Despite previously published data suggesting that organism detection is enhanced using PCR, concerns regarding the use of molecular testing to diagnose UTIs have been raised. These concerns include the possibility of identifying organisms that are not pathogenic in the presenting clinical episode, the lack of standardization of urinary molecular panels, and the costs associated with these novel tests. Our data suggest that these considerations are outweighed by the benefits for patients with cUTI; these patients have unique diagnostic and therapeutic challenges, and traditional UC may be an inadequate or insufficient diagnostic test. 

This group of patients typically presents to the urology office or emergency room with failed previous treatment or experience of frequent or rapid recurrences or progression due to (1) incorrect or insufficient therapy, or (2) failure to identify and address an underlying anatomic or comorbid condition. Our data clearly showed that UC poorly identifies UTIs caused by FOs and is inadequate in identifying all causative organisms present in PMOs, limiting the utility of UC in this population who are at risk for the presence of either or both. As the management of patients with cUTI presents a difficult clinical challenge, it is imperative that urologists and other specialists have access to diagnostic tools that provide potentially actionable information, especially for the correct identification of pathogenic organisms. The provider can subsequently assimilate these data along with other clinical information to determine the most appropriate treatment course. Importantly, while we found that PCR was superior to UC in overall organism detection, as well as detection of FOs and PMOs, our data do not support the abandonment of UC in this set of patients. Over 8% of UC+ specimens had potential pathogens not found by PCR, with nearly a quarter of these (1.9% of all UC+ organisms) not part of the PCR panel.

Our data are consistent with those of previous studies, in terms of the overall sensitivity advantage of PCR over UC. This analysis differs in that we specifically studied patients who met the clinical criteria for cUTI, and for this subset of patients, PCR provided potentially clinically actionable information that was not available for UC. Unique, potentially actionable pathogenic organisms were found in over 60% of PCR specimens. Specifically, PCR was 20 times more likely to identify PMOs, and nearly 36 times more likely to identify FOs. While both findings are not unexpected given the technological difference between UC and PCR, the frequency with which PMOs and FOs were found in cUTI and the magnitude of the difference between the tests were profound. With respect to collection methodology, specimen positivity appeared higher in both UC and PCR in patients whose specimens were collected via catheterization or clean catch; however, in neither case were the results significant. This may be due to the relatively small number of patients (1.9%) who had their specimens obtained via catheterization vs. via clean-catch methods. Interestingly, although it could be surmised that PMOs would be more prevalent on clean-catch vs. catheterized specimens due to enhanced detection of contaminants, we did not observe any difference in the detection of PMOs based on specimen collection technique in either UC or PCR, and the difference in PMO detection between the two tests was consistent across collection technique. This may be because our urology practice has established protocols for collection of midstream clean-catch urine specimens, and as a specialty practice, our staff is highly trained in communicating this technique to the patient.

This study has several limitations owing to its retrospective nature and technical factors associated with the tests themselves. Our practice guidelines dictate when urinary molecular testing *can* be performed, not that it *must* be performed. The determination of whether any specific diagnostic test is utilized, while broadly determined by clinical guidelines, is up to the discretion of the treating provider. As this study was a laboratory analysis that did not involve chart review, it is possible that there were some patients with cUTI that did not receive PCR testing; however, all patients who had PCR and UC performed were captured in this analysis. A fundamental limitation in the use of PCR is that its sensitivity is primarily limited by the predefined panel of organism-specific primers in microarray chips. This concern was offset using a multi-organism panel that covered a substantial number of pathogenic organisms in the urinary tract [14]. In addition, PCR can be affected by the presence of PCR inhibitors in the samples as well as technical issues, including low extraction efficiency and bubble formation in the OpenArray chip. These limitations were mitigated by utilizing a single, large-volume commercial laboratory with extensive experience in PCR testing. Conversely, the growth of organisms in UC may be affected by prior or recent antibiotic use, which may lead to exacerbating the apparent superiority of PCR. This limitation is mitigated as our purpose was specifically to compare PCR and UC in a subset of patients with cUTI—these patients often present having failed prior antibiotic therapy, and therefore our data illustrate the real-world challenges faced by clinicians managing this difficult-to-treat subset of patients. And while recent data suggest that genotypic antibiotic sensitivity detected by molecular techniques is similar to the phenotypic resistome [16], to ensure that clinicians were provided optimal data for patient management, our protocol calls for UC, PCR, and antibiotic resistance testing to be performed in conjunction on the same sample. Thus, while patients with cUTI may have been previously treated with antibiotics, the identification of a pathogen by culture and/or PCR and the associated antibiotic susceptibility and antibiotic resistance gene profile in a symptomatic patient is likely indicative of an active infection regardless of methodology. 

Although the detection of PMOs may be relevant in the management of cUTI, the presence of multiple organisms detected by PCR can be difficult to interpret. That said, the data strongly suggest that the presence of PMOs is associated with increased bacterial pathogenicity [17] as well as adverse outcomes in patients with cUTI [18]; therefore, identification of multiple pathogens in urine has prognostic value in addition to direct clinical utility. In addition, to ensure that the PCR report was meaningful, the reference laboratory utilized a cycle threshold limit for each organism that was defined as positive. With respect to FOs, the definition of an FO varies from laboratory to laboratory based on local technical considerations, particularly with respect to growth media, which was mitigated in this study as both PCR and UC testing were performed by the same commercial lab with consistent techniques during the study period; therefore, the data were internally consistent. Finally, this study did not include a review of antibiotic use or clinical courses on a patient–patient basis. As this was a comparative study on organism identification, chart review was not a component of this study; a prospective study to review the impact of PCR testing on patient care is in process.

This study did not include an analysis of next-generation sequencing (NGS) in this subset of patients as it is not, at present, a routine part of our clinical practice. Whole-genome shotgun sequencing is an excellent tool regarding unbiased pathogen detection in comparison to PCR, which has defined panels of pathogens to detect. While there is active research in the analysis of the urinary microbiome with over 350 publications per annum [19], publications on next-generation sequencing in UTI are scarcer, averaging fewer than 10 per year [20]. Indeed, the number of organisms identified by next-generation sequencing may be bewildering to the clinician—one analysis revealed a median 41 bacterial genera, 2 eukarya species, and 3 viral sequences *per sample* [21]. A recent meta-analysis comparing results of PCR and NGS found that PCR had a 99% sensitivity and a 94% specificity for diagnosing UTIs, while NGS was shown to have a 90% and 86% sensitivity and specificity, respectively, for identifying UTIs [22]. Future studies comparing PCR and NGS are needed to better define the role of these tests, but at present, there is limited need for whole-genome sequencing of clinical isolates for pathogen identification on a large scale given the availability of alternative technologies that offer a faster turnaround time, greater cost efficiency, and lower complexity. 

Finally, while this study was performed in a single independent specialty practice using a commercial laboratory, the management of cUTI and the prevalence of multi-drug-resistant organisms are international problems [23,24] that pose a substantial economic burden on individual institutions and healthcare systems [25,26]. The data suggest that this burden may be reduced by urinary molecular testing [27]; therefore, we believe that the findings are relevant not only to independent specialty practices, but to institutional settings in the United States and abroad as well.

## 5. Conclusions

Our data suggest that in the subset of patients with cUTI, PCR is significantly superior to UC in overall pathogen detection, and even more so in the identification of FOs and those with the presence of PMOs. PCR identified unique organisms not found in UC in most specimens, strongly supporting the premise that PCR has clinical validity and utility in providing treatment providers with potentially actionable data for patients with cUTI—these findings were independent of the method of specimen collection. Despite this, owing to limitations inherent in PCR testing, UC also identified organisms that were not identified by PCR, albeit in much smaller numbers. Using these tests in combination provides superior organism detection, thereby enhancing clinical decision making, which should improve outcomes, facilitate antibiotic stewardship, and ultimately reduce costs. We conclude that in this difficult-to-treat subset of patients, the addition of PCR to conventional UC is superior to either UC or PCR alone and should be incorporated into management pathways. 

## Figures and Tables

**Figure 1 biology-13-00257-f001:**
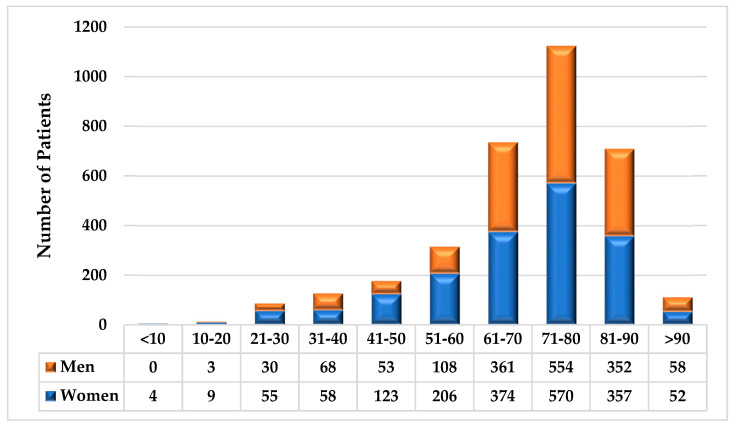
Patient Age and Gender Distribution.

**Figure 2 biology-13-00257-f002:**
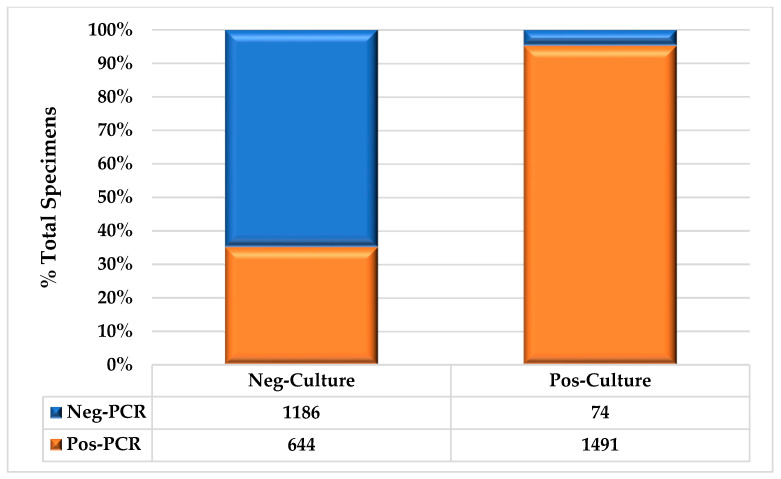
Test Results, UC vs. PCR.

**Table 1 biology-13-00257-t001:** Clinical Indications for Urinary Molecular Testing.

Recurrent UTI: ≥3/year or 2 in ≤6 months. Complicated UTI (any of the following): Male. Symptoms > 7 days. Comorbidities (systemic and GU): Diabetes mellitus. Immunosuppression. Neurogenic bladder. GU structural or functional abnormality. Urolithiasis. Recent hospitalization. GU instrumentation. Pyelonephritis.

**Table 2 biology-13-00257-t002:** List of 36 Organisms on UTI OpenArray Panel.

Organisms Included in PCR Panel		
*A. baumannii*	*E. cloacae*	*P. stuartii*
*A. urinae **	*E. faecalis*	*P. aeruginosa*
*A. omnicolens **	*E. faecium*	*S. marcescens*
*C. albicans*	*E. coli*	*S. aureus*
*C. glabrata*	*K. oxytoca*	*S. epidermidis*
*C. parapsilosis*	*K. pneumoniae*	*S. haemolyticus*
*C. amalonaticus*	*M. morganii*	*S. lugdunensis*
*C. freundii*	*M. tuberculosis*	*S. saprophyticus*
*C. koseri*	*M. genitalium **	*S. agalactiae*
*C. riegelii **	*M. hominis **	*S. pasteuranus*
*C. urealyticum **	*P. agglomerans*	*S. pyogenes*
*E. aerogenes*	*P. mirabilis*	*U. urealyticum **

Fastidious organisms are denoted by *.

**Table 3 biology-13-00257-t003:** List of 18 Antibiotic Resistance Genes on UTI OpenArray Panel.

ThermoFisher ABR Assay	ThermoFisher Assay ID *
VanA	Ba04646147_s1
β-lactamase (Class D)	Ba04646133_s1
AmpC β-lactamase (Class C)	Ba04646126_s1
Carbapenemase (Class A)	Ba04646151_s1
Carbapenemase (Class A)	Ba04646152_s1
Carbapenemase (Class B)	Ba04646131_s1
Carbapenemase (Class B)	Ba04931076_s1
Carbapenemase (Class D)	Ba04930816_s1
Extended-Spectrum β-lactamase (Class A)	Ba04646140_s1
Extended-Spectrum β-lactamase (Class A)	Ba04646153_s1
Extended-Spectrum-β-Lactamase	Ba04646149_s1
Extended-Spectrum-β-Lactamase	Ba04646142_s1
Extended-Spectrum-β-Lactamase	Ba04646154_s1
Extended-Spectrum-β-Lactamase	Ba04646127_s1
Macrolide Resistance	Pa04230913_s1
Quinolone Resistance	Ba04646160_s1
Quinolone Resistance	Ba04646145_s1
Verona integron-encoded metallo-β-lactamase	Ba04646155_s1

* Assay ID is provided to differentiate ABRs that have the same name.

**Table 4 biology-13-00257-t004:** PCR and UC Positivity, Catheterized vs. Clean-Catch Samples.

**PCR**	**Negative**	**Positive**	**Total**
Clean Catch	1243	2086	3329
Catheterized	17	49	66
Total	1260	2135	3395
**Urine Culture**	**Negative**	**Positive**	**Total**
Clean Catch	1802	1527	3329
Catheterized	28	38	66
Total	1830	1565	3395

**Table 5 biology-13-00257-t005:** Results of Corresponding Test in Positive Specimens by Collection Type.

**PCR-Positive**			
**Collection Method**	**UC-Negative**	**UC-Positive**	**Total**
Clean Catch	608	1442	2050
Catheterized	13	36	49
Total	621	1478	2099
**UC Positive**			
**Collection Method**	**PCR-Negative**	**PCR-Positive**	**Total**
Clean Catch	72	1455	1527
Catheterized	2	36	38
Total	74	1491	3395

**Table 6 biology-13-00257-t006:** Total organism counts in UC and PCR and organism line-item concordance.

Organism	PCR	UC	LIC	%PCR	%UC	*p*
*E. coli*	966	803	777	80.4%	96.8%	<0.01
* A. urinae *	416	9	8	1.9%	88.9%	<0.01
*S. pasteuranus*	373	0	0	0.0%	0.0%	n/a
*E. faecalis*	351	138	112	31.9%	81.2%	<0.01
*K. pneumoniae*	248	201	197	79.4%	98.0%	<0.01
*S. haemolyticus*	143	7	3	2.1%	42.9%	<0.01
*S. epidermidis*	129	63	57	44.2%	90.5%	<0.01
* U. urealyticum *	113	0	0	0.0%	0.0%	n/a
*P. aeruginosa*	94	60	56	59.6%	93.3%	<0.01
*S. Lugdunensis*	90	7	6	6.7%	85.7%	<0.01
*P. mirabilis*	71	50	44	62.0%	88.0%	<0.01
*S. agalactiae*	66	26	23	34.8%	88.5%	<0.01
*E. cloacae*	63	47	46	73.0%	97.9%	<0.01
* A. omnicolens *	61	0	0	0.0%	0.0%	n/a
* C. riegelii *	56	0	0	0.0%	0.0%	n/a
*E. faecium*	42	16	14	33.3%	87.5%	<0.01
*K. oxytoca*	42	34	28	66.7%	82.4%	0.12
*C. albicans*	34	2	2	5.9%	100.0%	<0.01
*C. glabrata*	33	0	0	0.0%	0.0%	n/a
*M. morganii*	30	10	9	30.0%	90.0%	<0.01
*S. aureus*	30	23	19	63.3%	82.6%	0.12
*E. aerogenes*	27	20	19	70.4%	95.0%	0.03
* M. hominis *	25	0	0	0.0%	0.0%	n/a
*C. koseri*	19	15	15	78.9%	100.0%	0.06
*S. marcescens*	15	7	6	40.0%	85.7%	0.04
*C. freundii*	13	15	4	30.8%	26.7%	0.81
*C. amalonaticus*	6	1	1	16.7%	100.0%	0.09
*S. saprophyticus*	5	5	5	100.0%	100.0%	n/a
*A. baumannii*	2	1	1	50.0%	100.0%	0.39
*P. stuartii*	2	1	1	50.0%	100.0%	0.39
*C. parapsilosis*	2	0	0	0.0%	0.0%	n/a
* M. genitalium *	2	0	0	0.0%	0.0%	n/a
*P. agglomerans*	1	1	1	100.0%	100.0%	n/a
*S. pyogenes*	1	0	0	0.0%	0.0%	n/a
*A. denitrificans **	0	1	0	0.0%	0.0%	n/a
*A. lwoffii **	0	1	0	0.0%	0.0%	n/a
*C. braakii **	0	2	0	0.0%	0.0%	n/a
*C. youngae **	0	1	0	0.0%	0.0%	n/a
*E. avium **	0	3	0	0.0%	0.0%	n/a
*P. hauseri **	0	3	0	0.0%	0.0%	n/a
*P. vulgaris **	0	2	0	0.0%	0.0%	n/a
*P. rettgeri **	0	3	0	0.0%	0.0%	n/a
*P. putida **	0	2	0	0.0%	0.0%	n/a
*R. ornithinolytica **	0	1	0	0.0%	0.0%	n/a
*R. planticola **	0	7	0	0.0%	0.0%	n/a
*S. liquefaciens **	0	1	0	0.0%	0.0%	n/a
* S. hominis * *	0	4	0	0.0%	0.0%	n/a
*S. simulans **	0	2	0	0.0%	0.0%	n/a
*S. warneri **	0	2	0	0.0%	0.0%	n/a
*S. xylosus **	0	1	0	0.0%	0.0%	n/a
Total	3571	1598	1454	40.7%	91.0%	<0.01

Fastidious organisms are highlighted in blue, and organisms not on the PCR panel are denoted by *.

**Table 7 biology-13-00257-t007:** Organisms Found on One Test Only (PMOs, Single Organisms, and Total).

	PMOs		Single Organisms		Total	
	UC	PCR	UC	PCR	UC	PCR
Found on One Test	19	877	128	456	147	1333
All Positive	33	896	1532	1239	1565	2135
%	57.6%	97.9%	8.4%	36.8%	9.4%	62.4%
*p*	<0.01		<0.01		<0.01	

## Data Availability

The original data in this study are restricted from public access due to patient privacy requirements.

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
