# Peer review of "Comparison of Polymerase Chain Reaction and Urine Culture in the Evaluation of Patients with Complex Urinary Tract Infections"

_biology, 2024, doi:10.3390/biology13040257_

Round 1

Reviewer 1 Report

Comments and Suggestions for Authors

Interesting way to detect contamination of wine. But the cost can be the matter. Good analyse for this topic. Hope to stand and develop a panel for frequent pathogens for those with CAKUT, and other co-morbidities. 

Additional comments:

1. What is the main question addressed by the research?

The main topic in debate is about the UC using classic method and PCR, an interesting one for rich country

2. What parts do you consider original or relevant for the field? What specific gap in the field does the paper address?

original is the basic comparison and the gap is linked with the categories of person involved in study

3. Please describe how the conclusions are or are not consistent with the evidence and arguments presented. Please also indicate if all main questions posed were addressed and by which specific experiments.

The main problem for a public hospital is if this comparison is one valid for today. in classic UC we have different methods that we use in the last 35 years, and PCR involve a lot of different training and equipment

Author Response

Kindly see the Microsoft Word attachment.  Line numbers refer to redlined document.

Reviewer 2 Report

Comments and Suggestions for Authors

There are few formatting errors e.g.105 should be 105. The retrospective nature of the study needs to be clarified. Is it an established practice of the institute to do PCR for all UTIs or only cUTI?  If yes, provide a justification of the practice. Provide a reference.

Additional comments are attached.

Author Response

(The authors gave the same response as above.)

Reviewer 3 Report

Comments and Suggestions for Authors

The study by Kapoor et al. on comparison between PCR detection of urinary bacteria and urine culture is interesting in view of the size of the material and concentrating on cUTI. Finding more bacteria by PCR than by culture is not new, as pointed out by the authors, not the least since UC was done aerobically and not looking for anaerobes or fastidious organisms in contrast to PCR.

I have some comments for the authors to consider:

1. A mid-stream urine sample has been shown less valid for detection of urinary pathogens both for UC and PCR. For the latter with higher sensitivity for detecting contaminants a clean-catch catheter sample is much better. We are not told about the differences between the two methods according to sample type, which would be a good idea.

2. For UC, a 1 mcl sample is very small, many labs use 10 mcl to detect <10-4 cfu/ml, because criteria for E.coli and S.saprophyticus are 10-3 cfu/ml. We are told, that positive samples by UC were 10-4 or higher, but was quantification of bacteria part of the PCR method? It is now well known, that the constantly positive bacteriuria or urinary microbiome as found by DNA methods usually varies around 10-1 to 10-2 cfu/ml, and clinical significance of these small number microbiota have not been proven. If data are avilable for quantification by PCR, these should be shown.

3. Regarding cUTI patients, many of these had probably been treated recently with antibiotics. This is another weakness of this study; UC will only detect living bacteria, while PCR can detect DNA from dead bacteria, and therefore present as better, but perhaps not more clinically relevant. Such data should be provided for the cases, if possible.

4. The ability to find PMO is not always clinically significant, since there is usually only one baterial pathogen causing infection; this may be different in catheterized patients, where several pathogens are usually found, but not necessarily treated, since removal of the catheter is the first option. This should be discussed.

5. PCR in this material did not reveal the antibiotic susceptibility of the organisms detected; this is a major weakness of this method. It could be paired with UC, so that important bacteria found by PCR were thereafter sought identified and AST´ed - but then the advantage  of speed wanes off.

6. There is no discussion on the obvious advantage of using whole genome sequencing (WGS) in stead of PCR. The quality of the latter depends on the primers chosen, as in this study, while with WGS all bacteria will be detected including antibiotic resistance genes and perhaps virulence genes. The price for WGS is becoming ever cheaper, which will probably mean, that PCR as in this study may soon become antiquated - both considering speed, bioinformatics and price.

Author Response

(The authors gave the same response as above.)

Round 2

Reviewer 3 Report

Comments and Suggestions for Authors

The authors have responded to the comments of the reviewer. No further comments